

# Five copper homeostasis gene clusters encode the Cu-efflux resistome of the highly copper-tolerant *Methylorubrum extorquens* AM1

Araceli Dávalos and Alejandro García-de los Santos

Programa de Ingeniería Genómica, Centro de Ciencias Genómicas, Universidad Nacional Autónoma de México, Cuernavaca, Morelos, Mexico

## ABSTRACT

**Background:** In the last decade, the use of copper has reemerged as a potential strategy to limit healthcare-associated infections and to control the spread of multidrug-resistant pathogens. Numerous environmental studies have proposed that most opportunistic pathogens have acquired antimicrobial resistance in their nonclinical primary habitat. Thus, it can be presumed that copper-resistant bacteria inhabiting a primary commensal niche might potentially colonize clinical environments and negatively affect the bactericidal efficacy of Cu-based treatments. The use of copper in agricultural fields is one of the most important sources of Cu pollution that may exert selection pressure for the increase of copper resistance in soil and plant-associated bacteria. To assess the emergence of copper-resistant bacteria in natural habitats, we surveyed a laboratory collection of bacterial strains belonging to the order *Rhizobiales*. This study proposes that *Methylorubrum extorquens* AM1 is an environmental isolate well adapted to thrive in copper-rich environments that could act as a reservoir of copper resistance genes.

**Methods:** The minimal inhibitory concentrations (MICs) of $CuCl_2$ were used to estimate the copper tolerance of eight plant-associated facultative diazotrophs (PAFD) and five pink-pigmented facultative methylotrophs (PPFM) belonging to the order *Rhizobiales* presumed to come from nonclinical and nonmetal-polluted natural habitats based on their reported source of isolation. Their sequenced genomes were used to infer the occurrence and diversity of Cu-ATPases and the copper efflux resistome of *Mr. extorquens* AM1.

**Results:** These bacteria exhibited minimal inhibitory concentrations (MICs) of $CuCl_2$ ranging between 0.020 and 1.9 mM. The presence of multiple and quite divergent Cu-ATPases per genome was a prevalent characteristic. The highest copper tolerance exhibited by *Mr. extorquens* AM1 (highest MIC of 1.9 mM) was similar to that found in the multimetal-resistant model bacterium *Cupriavidus metallidurans* CH34 and in clinical isolates of *Acinetobacter baumannii*. The genome-predicted copper efflux resistome of *Mr. extorquens* AM1 consists of five large (6.7 to 25.7 kb) Cu homeostasis gene clusters, three clusters share genes encoding Cu-ATPases, CusAB transporters, numerous CopZ chaperones, and enzymes involved in DNA transfer and persistence. The high copper tolerance and the presence of a complex Cu efflux resistome suggest the presence of relatively high copper tolerance in environmental isolates of *Mr. extorquens*.

Corresponding author
Alejandro García-de los Santos, alex@ccg.unam.mx

## INTRODUCTION

According to The World Health Organization (WHO), the increase in antibiotic-multiresistant bacteria and the limited therapeutic options to treat infections by these microorganisms are worldwide health problems that need urgent solutions (https://www.who.int/news-room/fact-sheets/detail/antimicrobial-resistance). In the last decade, the use of copper has reemerged as a potential antimicrobial agent in preventing healthcare-associated infections and controlling the spread of multidrug-resistant pathogens. The antimicrobial properties of ionic copper have been complemented with the "contact killing" approach using copper-coated surfaces to reduce bacterial transmission (*Grass, Rensing & Solioz, 2011*). The introduction of copper nanoparticles in textiles, latex, and other polymers used in healthcare environments has shown significant biocidal efficacy (*Arendsen, Thakar & Sultan, 2019*). Another promising therapeutic approach is the use of copper in combination or in complex with antibiotics, which increases its bactericidal efficacy (*Poole, 2017*).

The long-term success of these copper-based treatments requires information on the level of copper tolerance, the mechanisms conferring resistance, and its spread through horizontal gene transfer (HGT) not only among nosocomial pathogens but also within bacterial populations in their natural environments. This issue is particularly noteworthy since most opportunistic pathogens acquire multidrug resistance from their nonclinical primary habitat (*Samreen et al., 2021*; *Sanz-García et al., 2021*; *Thummeepak et al., 2020*; *Virieux-Petit et al., 2022*).

Whereas worldwide antibiotic resistance began in the mid-1950s with the excessive use of antibiotics for treating infectious diseases (*Hutchings, Truman & Wilkinson, 2019*), the selection pressure exerted by copper started two billion years ago, during the great oxidation event that increased the bioavailability of copper. Since then, bacteria have evolved different mechanisms to maintain intracellular copper at trace concentrations and avoid toxicity (*Borkow & Gabbay, 2009*; *Coombs & Barkay, 2005*; *Dupont, Grass & Rensing, 2011*). Copper extrusion from the cytoplasm is the most prevalent and crucial mechanism for maintaining the homeostasis of copper (Fig. S1). This copper translocation is performed mainly by inner membrane $P_{1B-1}$-type ATPases (Cu-ATPases) also known as CopA (*Argüello, Raimunda & Padilla-Benavides, 2013*). In some bacteria, in addition to Cu-ATPases, the extrusion of copper from the cytoplasm to the periplasm requires two Cu-chaperones CopZ and CusF. CopZ is a cytoplasmic protein that binds $Cu^+$ and delivers it to Cu-ATPases. Later, Cu-ATPases transfer $Cu^+$ to the periplasmic Cu-chaperon CusF. Finally, CusF delivers $Cu^+$ to the proton-cation antiporter (RND) CusCBA a three-protein transport system that pumps $Cu^+$ out of the cell (*Giachino & Waldron, 2020*). In *Rhizobium etli* and *R. tropici* the disruption of Cu-ATPases-encoding genes produces copper-sensitive mutants indicating that these copper-efflux pumps are the core

component of the homeostasis system (*Elizalde-Díaz et al., 2019*; *González-Sánchez et al., 2018*). Homologous cytoplasmic and periplasmic CopZ and CusF are poorly conserved and the presence of a CusCBA system has not been demonstrated (*Cubillas et al., 2017*).

In the last two centuries, the natural dissemination of copper through geochemical cycles was surpassed by anthropogenic industrial and agricultural activities that have polluted air, soil, and aquatic environments (*Elguindi et al., 2011*; *Vardhan, Kumar & Panda, 2019*). The use of copper-based pesticides, insecticides and fertilizers is a very important source of pollution in agricultural fields (*Briffa, Sinagra & Blundell, 2020*). This uncontrolled copper pollution has been presumed to exert a progressive selection pressure increasing the diversity and dispersion of copper resistance genes (*Shaw et al., 2020*). This overlooked resistance would negatively affect the effectiveness of copper-based therapies. Studies in *Acinetobacter baumannii* revealed the presence of highly conserved copper-resistance genes in clinical, wastewater, and nonpolluted environmental isolates (*Irawati et al., 2021*; *Thummeepak et al., 2020*; *Williams et al., 2016*).

There are few studies on copper tolerance in bacteria of agricultural relevance. The order *Rhizobiales* (α-Proteobacteria) is a cosmopolitan group of bacteria that include plant-associated facultative diazotrophs (PAFD) that can live freely in the rhizosphere or as intracellular symbiotic nitrogen-fixing legume-nodulating bacteria (*Wang et al., 2020*). This order also includes pink-pigmented facultative methylotrophs (PPFM) that can use one-carbon compounds (C1, lacking C-C bonds), such as methanol, as the sole source of carbon and energy. The genera *Methylobacterium* (*Mb*) and *Methylorubrum* (*Mr*) have a cosmopolitan lifestyle and inhabit plant leaves and stems, aquatic sediments, dust, air, water, and soil. Plant-associated species contribute to plant growth producing phytohormones, secondary metabolites, and several nitrogen-fixing species supply ammonia (*Dourado et al., 2015*; *Green & Ardley, 2018*). The copper tolerance of species potentially exposed to copper-based pesticides and fertilizers in agricultural fields has not been reported. Several species with resistance to multiple antibiotics have been found in municipal water distribution systems or hospital tap water (*Furuhata et al., 2006*; *Gallego, García & Ventosa, 2006*). Their chlorine resistance, biofilm formation, desiccation tolerance, and high-temperature resistance have raised concerns about their potential risk as emerging pathogens (*Szwetkowski & Falkinham, 2020*). Such concerns are supported by several reports indicating that multidrug-resistant *Methylobacterium* species are opportunistic pathogens that cause infections in immunocompromised patients (*Cordovana et al., 2019*; *Kovaleva, Degener & van der Mei, 2014*; *Lai et al., 2011*). Despite its medical relevance, there is no information available on the copper tolerance of PPFM isolated from clinical and nonmetal-polluted environments.

To assess the emergence of copper-resistant bacteria in natural habitats, in this study, we present a functional and genomic analysis of copper tolerance in eight plant-associated facultative diazotrophs (PAFD), and five pink-pigmented facultative methylotrophs (PPFM) belonging to the order *Rhizobiales* maintained in a laboratory collection and presumed to come from nonclinical and nonmetal-polluted natural habitats based on

**Table 1 Strains, sources or hosts, and genome structure.**

| Strains[a] | Abbreviated name | Source/Host | Reference | Number of replicons[b] | | |
|---|---|---|---|---|---|---|
| PAFDs | | | | Chro | Mp | P |
| *Rhizobium etli* CFN42[T] | Ret | Nodules of bean plants, México. | *Segovia, Young & Martínez-Romero (1993)* | 1 | 0 | 6 |
| *Rhizobium tropici* CIAT899[T] | Rtr | Nodules of bean plants, Colombia. | *Martínez-Romero et al. (1991)* | 1 | 1 | 1 |
| *Rhizobium freirei* PRF81[T] | Rfr | Nodules of bean plants, Brazil. | *Dall'Agnol et al. (2013)* | 1 | 0 | 2 |
| *Rhizobium leguminosarum* bv viciae 3841 | Rle | UK/Peas and broad beans | *Johnston & Beringer (1974)* | 1 | 0 | 7 |
| *Sinorhizobium meliloti* 1021 | Sme | Australia/nodules of Alfalfa | *Galibert et al. (2001)* | 1 | 2 | 0 |
| *Sinorhizobium fredii* NGR234 | SfrNGR | Nodules of *Lablab purpureus*, New Guinea | *Trlnlck (1980)* | 1 | 1 | 1 |
| *Sinorhizobium fredii* GR64 | SfGR | Nodules of bean plants, Spain. | *Herrera-Cervera et al. (2006)* | 1 | 0 | 2 |
| *Mesorhizobium loti* MAFF303099 | Mlo | Nodules of *Lotus japonicum*, Japan | *Saeki & Kouchi (2000)* | 1 | 0 | 2 |
| PPFMs | | | | | | |
| *Methylorubrum extorquens* AM1 | MeAM | Airborne contaminant, UK. | *Peel & Quayle (1961)* | 1 | 1 | 3 |
| *Methylorubrum extorquens* TK0001[T] | MexTK | Soil, Poland. | *Urakamit & Komagata (1984)* | 1 | 0 | 0 |
| *Methylobacterium radiotolerans* JCM2831[T] | Mra | Rice-grains Gamma- irradiated, Japan. | *Ito & Iizuka (1971)* | 1 | 0 | 8 |
| *Methylobacterium organophilum* DSM760[T] | Mor | Lake bottom mud, USA. | *Patt, Cole & Hanson (1976)* | 1 | 0 | 0 |
| *Methylobacterium nodulans* ORS2060[T] | Mno | Nodules of *Crotalaria podocarpa*, Senegal. | *Jourand et al. (2004)* | 1 | 0 | 7 |

**Notes:**
[a] Type strains according with the list of prokaryotic names with standing nomenclature (LPSN; *Parte et al., 2020*).
[b] Chro, chromosome; Mp, megaplasmids (size >1,000 kb); P, plasmid (size <1,000 kb).

reports on their source of isolation (Table 1). Their copper tolerance varies between 0.020 and 1.9 mM. PPFM exhibit higher MICs than PAFD. The presence of multiple and quite divergent Cu-ATPases per genome is a predominant characteristic in both bacterial groups. The genome-predicted copper efflux resistome of *Mr. extorquens* AM1, the strain with the highest MIC to $CuCl_2$ (1.9 mM), consists of five large (6.7 to 25.7 kb) Cu homeostasis gene clusters (Cu-HGCs), three clusters shared genes encoding Cu-ATPases, CusAB transporters, numerous CopZ chaperones, and enzymes involved in DNA transfer and persistence. These findings provide clues on the emergence of a bacterial reservoir that may contribute to disseminating antimicrobial resistance genes.

## MATERIALS AND METHODS

### Relevant characteristics of PAFD and PPFM

The strains analyzed in this work (Table 1) are model PAFD and PPFM, presumed to come from niches not polluted with metals, widely used in laboratory and field studies. All the strains were kindly donated by Professor Esperanza Martínez Romero (Centro de Ciencias Genómicas, UNAM, México City, México), and were carefully maintained in 20% glycerol stored at −70 °C. All the PAFD were isolated from the inside of root nodules of different species of leguminous plants growing in a variety of habitats in different countries

(*Dall'Agnol et al., 2013*; *Galibert et al., 2001*; *Herrera-Cervera et al., 2006*; *Johnston & Beringer, 1974*; *Martínez-Romero et al., 1991*; *Saeki & Kouchi, 2000*; *Segovia, Young & Martínez-Romero, 1993*; *Trlnlck, 1980*). The studied PPFM are cosmopolitan freestyle bacteria isolated from different environments around the world. This group includes two *Mr. extorquens* strains and three *Mb.* species. *Mb. nodulans* ORS2060$^{\text{T}}$ is a facultative methylotroph and diazotroph isolated from the inside of root nodules of *Crotalaria podocarpa* in Senegal. Although it is a non-pigmented species, to simplify the results, it was included in the PPFM group (*Ito & Iizuka, 1971*; *Jourand et al., 2004*; *Patt, Cole & Hanson, 1976*; *Peel & Quayle, 1961*; *Urakamit & Komagata, 1984*). The ability of these bacteria to grow from methanol as the sole carbon source and the pink-pigmented phenotype were verified in this study. The sequenced genomes of PAFD and PPFM can be freely accessed through NCBI genome resources (https://www.ncbi.nlm.nih.gov/home/genomes/). A list of accession numbers is included in Table S1.

## Media cultures

Bacterial cells maintained at −70 °C were propagated in peptone-yeast (PY)-agar medium containing 5 g/L peptone, 3 g/L yeast extract, and 15 g/L agar. After sterilization, 10 ml/L 0.7 M $CaCl_2$ was added.

Minimal medium (Mm) is a chemically defined medium prepared from three solutions (A, B, and C) and sterilized separately. Solution A contained 1.620 g/L sodium succinate hexahydrate as a carbon source, 0.534 g/L $NH_4Cl$ as a nitrogen source, 0.219 g/L $K_2HPO_4$, and 0.1 g/L $MgSO_4$, and its pH was adjusted to 6.8 before sterilization. Agar (15 g/L) was added and then sterilized in an autoclave. Solution B contained filter-sterilized 0.025 g/5 ml $FeCl_3 \cdot 6H_2O$, and solution C contained 0.7 M $CaCl_2 \cdot 2H_2O$ (autoclaved). One milliliter of B solution and two milliliters of C solution were added to one liter of previously sterilized A solution.

## Estimation of minimal inhibitory concentrations (MICs) of $CuCl_2$

We followed the definition and standardized dilution method set by the British Society of Antimicrobial Chemotherapy and updated by *Andrews (2001)*, and by the European Committee for Antimicrobial Susceptibility Testing (*EUCAST, 2000*) to estimate the MICs of an antimicrobial agent. Based on these studies, the MIC for each strain was defined as the lowest concentration of $CuCl_2$ that consistently prevented visible growth in at least three independent assays. Based on this definition, our raw data were images of a solid medium with increasing millimolar (mM) concentrations of $CuCl_2$ with or without the growth of resistant or susceptible strains. Representative examples of growth monitored photographically at 2, 4, and 6 days post-propagation are shown in Fig. S2. In this example, the PPFM *M. sp.* able to grow in 2 mM $CuCl_2$ was excluded from this study because its genome sequence is not available. Numerous images were used to construct a table of permissive and inhibitory concentrations of $CuCl_2$ for each strain (Table S2). The workflow used to assess MICs is shown in Fig. S3.

## Medium, CuCl$_2$ solutions, inoculation of plates, and reproducibility of MICs

To maintain the reproducibility of MICs in different assays we used agar plates with chemically defined medium instead of solid PY-rich medium. The variation observed in PY plates may be due to the high binding affinity of Cu(I) for amino acids.

To maintain repeatability among assays, the range of CuCl$_2$ concentrations evaluated (0–2 mM) was added to the mm from a 50 mM stock solution of CuCl$_2$-2H$_2$O (Sigma–Aldrich, St Louis, MO, USA) prepared in Milli–Q water and filter sterilized. The stock solution of CuCl$_2$-2H$_2$O was used on the day of preparation and then discarded.

Preparation of the inoculum, in the same way, was also important to avoid variation among MIC assays. Each strain was propagated in solid PY medium. After 3 days at 30 °C, the cultured bacteria were inoculated into 3 ml of PY broth. After overnight growth, cultures were adjusted to OD$_{620}$ = 0.3, washed twice with 10 mM MgSO4, and tenfold serially diluted ($10^{-1}$–$10^{-6}$). Twenty microliters from each dilution were spotted on solid Mm supplemented with increasing millimolar (mM) concentrations of CuCl$_2$.

## Occurrence of copper translocating P$_{1B}$-type ATPases (Cu-ATPases) encoded in the genomes of PAFD and PPFM

Cu-ATPases encoded in the PAFD and PPFM genomes were searched by BLASTP (https://blast.ncbi.nlm.nih.gov/Blast.cgi). The sole Cu-ATPase encoded in the genome of *R. etli* CFN42 was used (WP_011427866.1) as a query protein, because of functional evidence previously demonstrated by the copper sensitivity phenotype of a mutant *actP::Ω* Sp (González-Sánchez et al., 2018). Cu-ATPases containing eight transmembrane helices (TMHs) typical for Cu-ATPases were filtered from metal-ATPases using TOPCONS (http://topcons.net/). Additionally, we used CD-Search (https://www.ncbi.nlm.nih.gov/Structure/cdd/wrpsb.cgi) to verify the presence of the P-type ATPase Cu-like domain cd02094 (NCBI) containing three invariant amino acid signature motifs involved in Cu(I) binding and translocation: two cysteine residues CXC in TMH6; one tyrosine, one asparagine, and one proline YN(X$_4$)P residue in TMH7; and one methionine followed by serine residue MXXSS in TMH8 (Arguello, 2003). The multiple sequence alignment of Cu-ATPases performed with Clustal Omega version 1.2.4 at MBL-EBI (https://www.ebi.ac.uk/Tools/msa/clustalo/) shows the signature motifs shared in 28 Cu-ATPases (Fig. S4). To verify that we did not miss Cu-ATPases by BLASTP searches, we retrieved all the metal translocating P-type ATPases annotated in the target genomes from the Bacterial and Viral Bioinformatics Resource Center (https://www.bv-brc.org/) formerly Pathosystems Resources Integration Center (https://patricbrc.org). The Cu-ATPases were filtered by searching the signature motifs with CD-Search. The occurrence analysis is summarized in Table 2. The 28 amino acid sequences obtained are shown in the Table S3 (Cu-ATPase sequences).

Table 2 Occurrence of Cu-ATPases in the genomes of PAFDs and PPFMs with different $CuCl_2$ tolerance.

| PAFDs | MIC $CuCl_2$(mM) | Number of Cu-ATPases | Genome location | NCBI RefSeq |
|---|---|---|---|---|
| *Rhizobium freirei* PRF81 | 0.100 | Rfr1Ch | Chromosome | WP_037155004.1 |
| | | Rfr2Ch | Chromosome | WP_004111316.1 |
| *Sinorhizobium fredii* NGR234 | 0.090 | SfrNGR1Mpb | MpNGR234b | ACP22182.1 |
| | | SfrNGR2Mpb | | ACP22198.1 |
| | | SfrNGR3Mpb | | ACP22696.1 |
| *Sinorhizobium meliloti* 1021 | 0.030 | Sme1Mpa | MpSma | WP_162471698.1 |
| | | Sme2Mpa | | WP_010967529.1 |
| | | Sme3Mpb | MpSmb | WP_010975861.1 |
| *Mesorhizobium loti* MAFF303099 | 0.025 | MloCh | Chromosome | WP_010913136.1 |
| *Rhizobium tropici* CIAT899 | 0.025 | RtrCh | Chromosome | WP_015341555.1 |
| *Sinorhizobium fredii* GR64 | 0.02 | SfrGR1 | Draft genome | WP_192817378.1 |
| | | SfrGR2 | | WP_040959567.1 |
| *Rhizobium leguminosarum* bv viciae 3841 | 0.02 | Rle1Ch | Chromosome | WP_011651514.1 |
| | | Rle2Pl11 | pRL11 | WP_011654772.1 |
| | | Rle3Pl11 | | WP_011655075.1 |
| *Rhizobium etli* CFN42 | 0.02 | Retpe | pCFN42e | WP_011427866.1 |
| Total Cu-ATPases | | 5 | Chromosome | |
| | | 9 | Mp/p | |

| PPFMs | MIC $CuCl_2$(mM) | Number of Cu-ATPases | Genome location | NCBI RefSeq |
|---|---|---|---|---|
| *Methylorubrum extorquens* AM1 | 1.9 | MeAM1Ch/CopA1 | Chromosome | WP_012753106.1 |
| | | MeAM2Ch/CopA2 | | WP_158022369.1 |
| | | MeAM3Ch/CopA3 | | WP_238231531.1 |
| | | MeAM4Mp/CopA4 | Mp | ACS43114.1 |
| | | MeAM5Mp/CopA5 | | ACS43019.1 |
| *Methylobacterium organophilum* DSM760 | 0.800 | Mor1Ch | Chromosome | PVZ04141.1 |
| | | Mor2Ch | | PVZ07231.1 |
| *Methylobacterium radiotolerans* JCM2831 | 0.150 | Mra1Ch | Chromosome | WP_012320523.1 |
| *Methylobacterium nodulans* ORS2060 | 0.125 | Mno1Ch | Chromosome | ACL55831.1 |
| | | Mno2p2 | pMNOD02 | WP_012631078.1 |
| | | Mno3p2 | | WP_012631108.1 |
| *Methylorubrum extorquens* TK0001 | 0.075 | MexTK1Ch | Chromosome | WP_101475842.1 |
| Total Cu-ATPases | | 7 | Chromosome | |
| | | 4 | Mp/p | |

**Note:**
Genome location indicates occurrence in chromosomes, megaplasmids (Mp), and plasmids (p).

## Evolutionary divergence among Cu-ATPases encoded in the genomes of PAFD and PPFM

The 28 amino acid sequences, obtained as described above, were used to infer their phylogeny. Four sequences were used as an outgroup: two putative zinc transporters encoded in the genomes of *Mr. extorquens* AM1 (WP_012752954.1) and *Mb. organophilum* DSM760 (PVZ05212.1), as well as two Cu-ATPases encoded in the genome

of *Acidithiobacillus ferrivorans* ACH (WP_215894663.1) and (WP_215895207.1). Multiple sequence alignment, phylogenetic reconstruction analyses, and visualizations of phylogenetic data were performed using the pipeline of ETE3 at GenomeNet (https://www.genome.jp/tools/ete/). The workflow was as follows: clustalo_default-trimal001-prottest_default-phyml_default_boostrap. Branch support was computed using 100 bootstrapped trees. The Cu-ATPase sequences used to infer the phylogeny are available in Table S4 (Cu-ATPases for phylogeny).

## Contextual information from conserved gene neighborhood (GN) analysis

This analysis was performed using TREND (*Gumerov & Zhulin, 2021*), a freely available bioinformatic tool for the tree-based exploration of neighborhoods and domains (http://trend.evobionet.com/). The five Cu-ATPases (CopA1-CopA5) encoded in the genome of *Mr. extorquens* AM1 (Table S5, CopA sequences for TREND) were used as input information in the neighborhoods pipeline. As a result, we obtained a phylogenetic tree combined with interactive gene neighborhoods that showed predicted operons based on the distance between genes, provided domains and features of gene products, and identified homologous proteins. Unexpectedly, the *copA2*. gene neighborhoods could not be obtained using the neighborhoods pipeline of TREND. These data were manually obtained at NCBI using CopA2 NCBI Refseq as input (WP_158022369.1) (see the workflow in Fig. S5).

## Inference of the genome-based Cu efflux resistome in the PPFM *Mr. extorquens* AM1

The thoughtful curation of the contextual information obtained for the five *copA* genes was used to infer five different Cu-Homeostasis Gene Clusters (Cu-HGCs). Their gene organization is listed in Table S6, (Cu-HGCs). Their potential mobility was inferred by the presence of proteins whose predicted function has been associated with mobile genetic elements (MGEs) and horizontal gene transfer (HGT). The proteins encoded in each Cu-HGC are listed in Table S6, (Cu-HGCs).

# RESULTS

## Comparison of CuCl$_2$ MICs between PAFD and PPFM revealed hypertolerance of *Mr. extorquens* AM1

Based on the MIC of copper reported for bacteria isolated from mineral deposits, polluted environments and clinical samples we defined the range of high tolerance between 1- and 20-mM (*Cusick et al., 2021*; *Watkin et al., 2009*; *Yik et al., 2018*). In the present study, the MIC for each strain was obtained as described in the Material and Methods. Figure 1 shows the MIC of CuCl$_2$ that consistently prevented visible growth of the PAFD and PPFM strains in three independent assays. Figures 1A and 1B show that PPFM are more tolerant to CuCl$_2$ than PAFD. The highest CuCl$_2$ MIC observed in *Mr. extorquens* AM1 (1.9 mM) was similar to the MICs reported for bacterial strains isolated from mines,

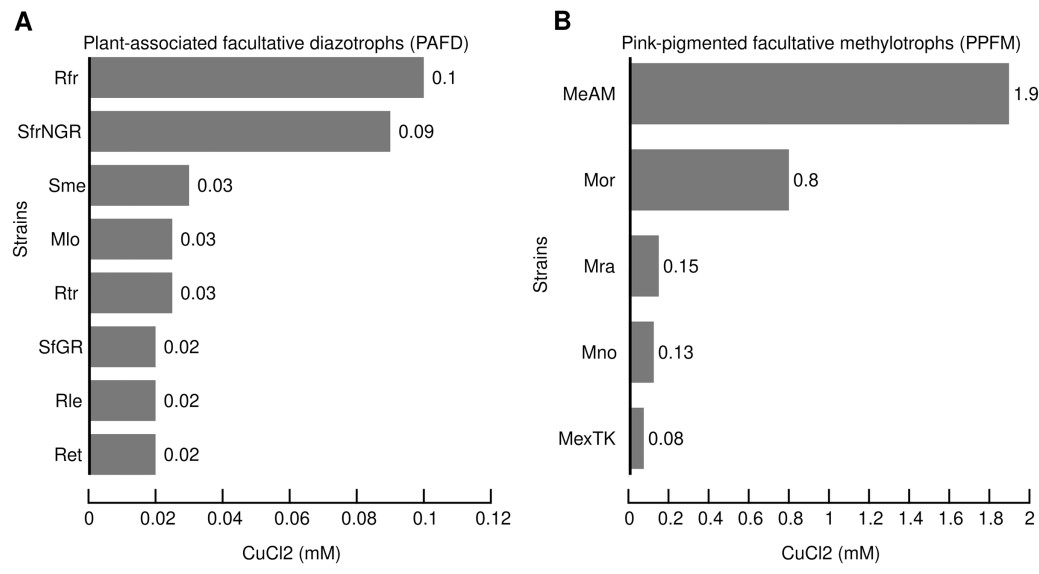

**Figure 1** **Minimal inhibitory concentrations (MICs) of CuCl$_2$ (mM) that consistently inhibit the visible growth of PAFD (A) and PPFM (B) on solid chemical defined medium.** MICs were assessed as detailed in Methods section. Names of PAFD strains: Rfr, *R. freirei* PRF81; SfrNGR *S. fredii* NGR234; Sme, *S. meliloti* 1021; *M. loti* MAFF303099, *R. tropici* CIAT899; *S. fredii* GR64, Rle, *R. leguminosarum* bv viciae 3841; *R. etli* CFN42. Names of PPFM: MeAM, *Mr. extorquens* AM1; Mor, *Mb. organophilum* DSM760; *Mb. radiotolerans* JCM2831; *Mb. nodulans* ORS2060; *Mr. extorquens* TK0001.

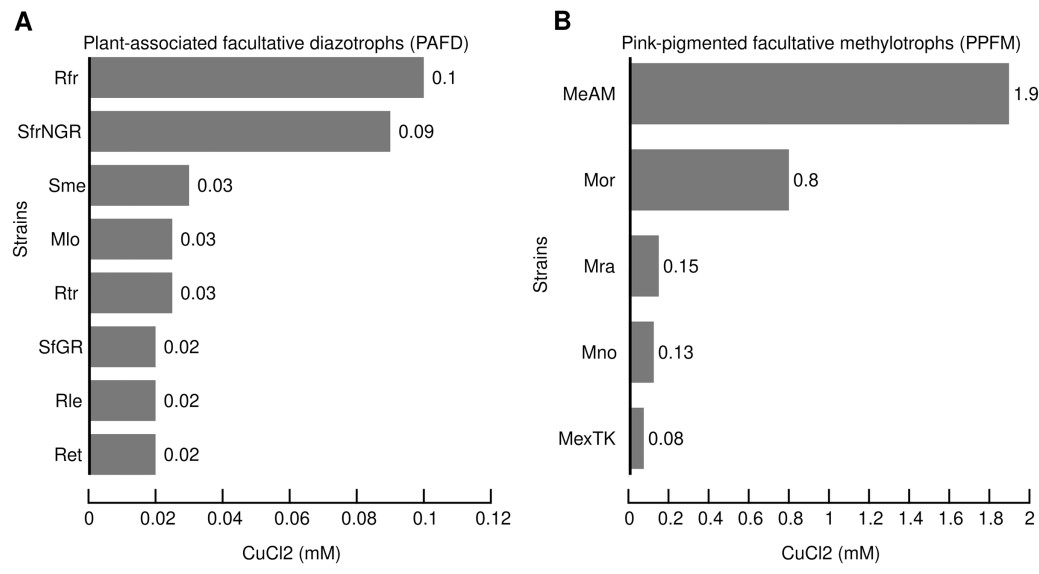

industrial wastewater, and clinical samples, such as *S. meliloti* CCNWSX0020 (1.8 mM), *C. metallidurans* CH34 (2 mM), and *A. baumannii* (1.5 mM) (*Li et al., 2014*; *Wiesemann et al., 2013*; *Williams et al., 2016*).

## Several PAFD and PPFM harbor multiple Cu-ATPases

Copper extrusion from the cytoplasm to the periplasm is the most prevalent and crucial mechanism for maintaining the homeostasis of copper. It is performed mainly by P$_{1B-1}$-type ATPases known as CopA (*Argüello, Raimunda & Padilla-Benavides, 2013*). Since the presence of multiple CopA is a characteristic highly conserved in the genome of *Rhizobiales* (*Cubillas et al., 2017*), we performed an occurrence analysis to investigate the possibility of a correlation between the number of Cu-ATPases in the genomes of PAFD and PPFM and their MICs. Table 2 indicates that the number of Cu-ATPases per genome varied from 1 to 5, and their role in copper translocation was supported by the presence of amino acid motifs CXC, YN(X$_4$)P, and M(X$_3$)S located at transmembrane helices 6, 7, and 8, respectively (Fig. S4). Although the highest MIC of *Mr. extorquens* AM1 (1.9 mM) correlated with its highest number of Cu-ATPases (5), no clear correlation could be established between MICs and the number of Cu-ATPases in other species.

## Cu-ATPases from PAFD and PPFM show high genetic diversity

Table 2 shows that multiple Cu-ATPases are encoded in different replicons, namely, chromosomes, megaplasmids, and plasmids. This result suggests that genes encoding Cu-ATPases may have different evolutionary histories. To test this hypothesis, the

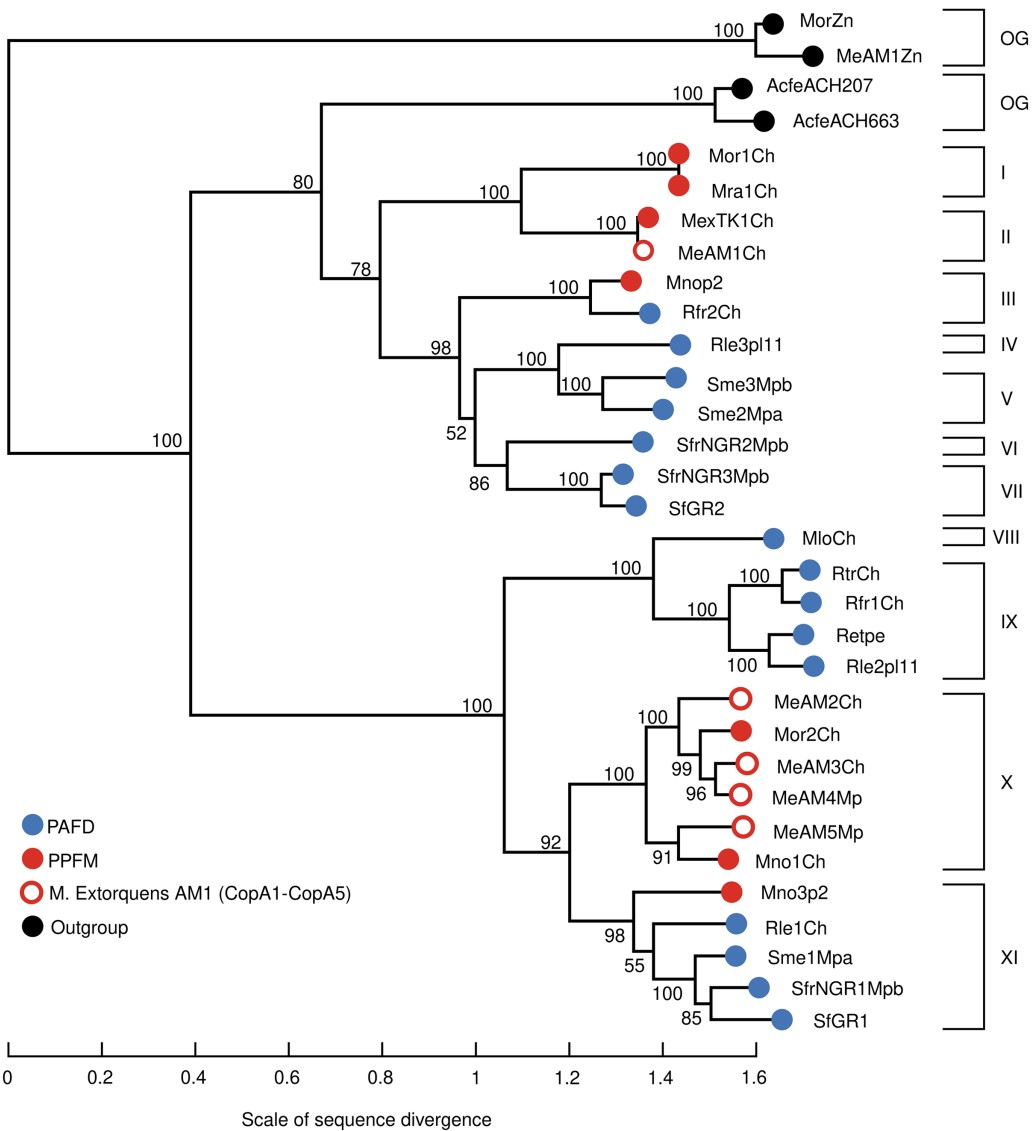

**Figure 2 Maximum likelihood phylogenetic tree inferred from amino acid sequences of Cu-ATPases encoded in the genomes of PAFD and PPFM listed in Table 2.** Copper translocating ATPases from *Acidithiobacillus ferrivorans* ACH (AcfeACH207) and (AcfeACH663,) as well as Zn-ATPases from *Mb. organophilum* DSM760 (MorZn) and *Mr. extorquens* AM1 (MeAM1Zn) were included as outgroup (OG). Cluster support was assessed by Bootstrap with 100 replicates, values are indicated at nodes. The scale of genetic change is indicated at the bottom. Cu-ATPases clustered in the same lineages share similarity values >85%. Cu-ATPases grouped in different lineages share similarity values <85%. Genome and protein names are in Table 2.

evolutionary divergence of the 28 Cu-ATPases (Table S4 Cu-ATPases for phylogeny) was inferred by maximum likelihood phylogenetic analysis. The 28 Cu-ATPases from PAFD and PPFM were sorted into eleven distantly related lineages (Fig. 2). Large divergence was also observed among the multiple Cu-ATPases encoded in the genome of several species, suggesting evolution by HGT. The phylogeny also showed that the Cu-ATPases MeAM3Ch and MeAM4Mp from *Mr. extorquens* AM1 located in lineage X were closely related, suggesting evolution by gene duplication.

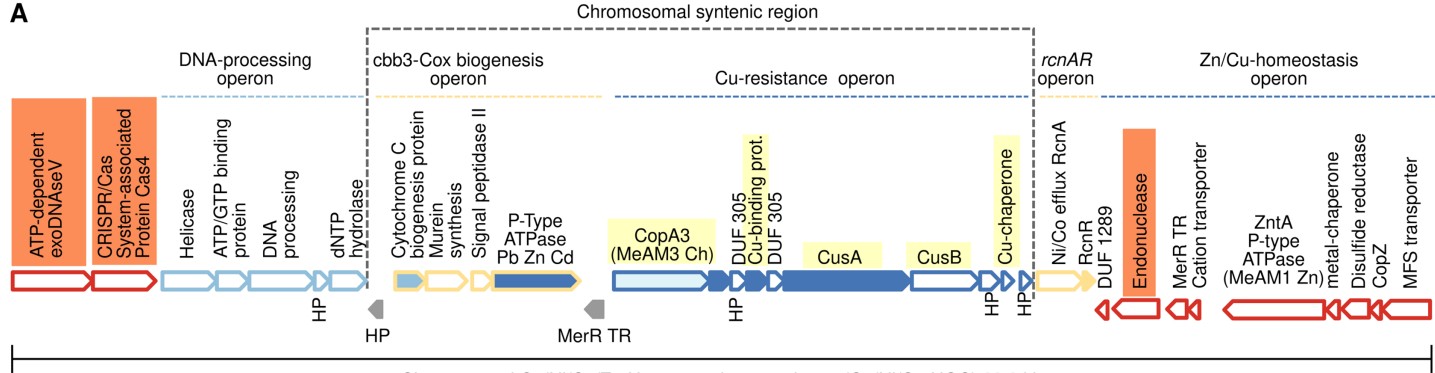

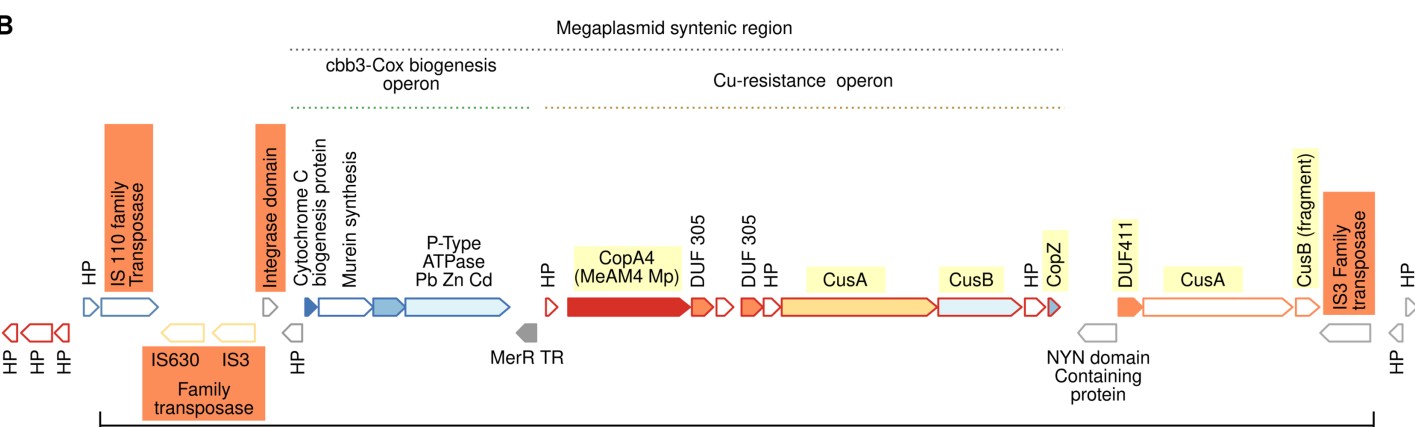

**Figure 3 Schematic comparison of the multimetal Cu/Ni/Co/Zn-HGC (A) and the Cu-HGC-1 (B) encoded in the chromosome and megaplasmid, respectively of *Mr. extorquens* AM1.** Gene products are indicated by arrow symbols. Proteins with conserved domains are fully colored; homologs are in same colors; bold border indicates protein used as query. Genes belonging to the same operon have border of the same color. Genes encoding hypothetical proteins and transcription regulators proteins are indicated with HP and TR respectively. The co-occurrence of different Cu-efflux systems (Cu-ATPases, CusAB and Cu-chaperones) are highlighted in yellow. Syntenic regions shared between Cu/Ni/Co-HGC and Cu-HGC-1 are indicated with dashed lines. The co-occurrence of different Cu-efflux systems (Cu-ATPases, CusAB and Cu-chaperones) are highlighted in yellow.

### Gene neighborhoods analysis of Cu-ATPases uncovers a complex and potentially mobile Cu efflux resistome in *Mr. extorquens* AM1

The high copper resistance exhibited by *Mr. extorquens* AM1 and the presence of five Cu-ATPases (CopA1-CopA5) encoded in its genome led us to define the Cu efflux resistome based on contextual information of *copA1-copA5* genes as detailed above in Materials and Methods. This contextual information revealed that the five CopA belong to five different Cu-Homeostasis Gene Clusters (Cu-HGCs). Detailed schemes for each Cu-HGC, based on the interactive gene neighborhood diagrams displayed in TREND, are shown in Figs. 3–5.

The proteins encoded in each Cu-HGC are listed in Table S6 (Cu-HGCs). The organization of the Cu efflux resistome in five Cu-HGCs is described below.

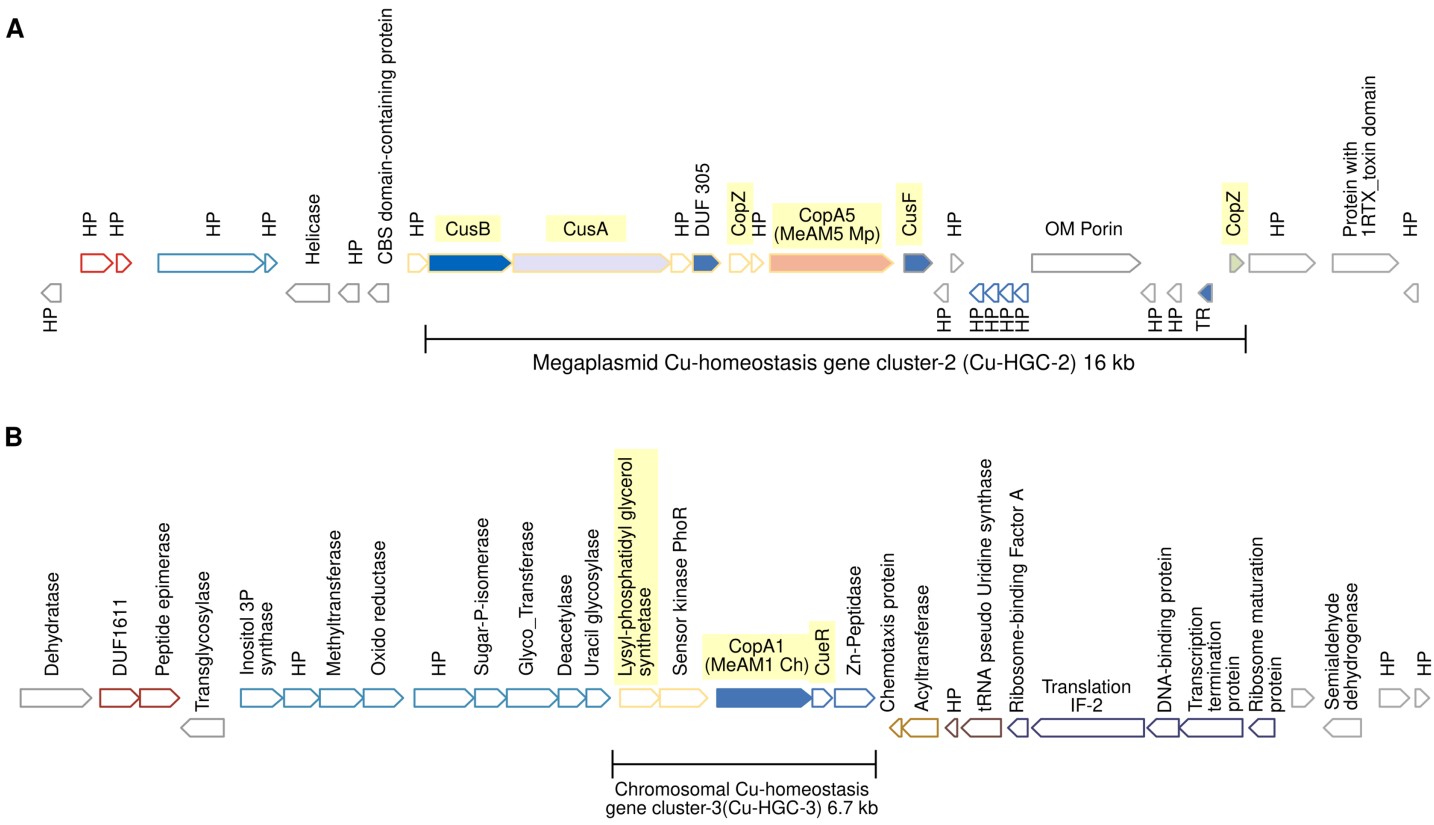

**Figure 4 Schematic view of the genomic organization of Cu-HGC-2 and Cu-HGC-3.** (A) The Cu-HGC-2 is characterized by the co-occurrence of Cu-ATPases, CusAB-RND transporters and Cu-chaperones (highlighted in yellow). (B) The Cu-HGC-3 shows the co-occurrence of genes encoding CopA1 and a phosphatidylglycerol synthetase (highlighted in yellow) that may confer Cu resistance under low pH (see Results section).

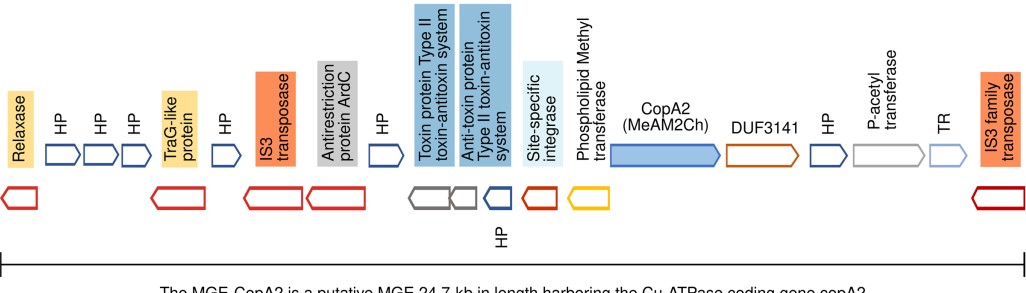

**Figure 5 Genomic organization of MGE-CopA2.** The Cu-ATPase CopA2 is harbored in a putative mobile genetic element containing a repertoire of proteins frequently found in mobile genetic elements (highlighted in colors) and IS transposases flanking both sides. This genomic organization was named MGE-CopA2.

## Chromosomal 23.9 kb Cu/Ni/Co-homeostasis gene cluster (Cu/Ni/Co-HGC)

We first analyzed the GN of the Cu-ATPase coding gene *copA3* due to the close phylogenetic relationship with CopA4, which suggests a gene duplication event (Fig. 2, lineage X). A careful inspection revealed that *copA3* is part of a putative Cu-resistance operon (Fig. 3A, blue dashed line). Three additional metal resistance operons were predicted, with one upstream (light yellow dashed line) and two downstream of the Cu-resistance operon (Fig. 3A, dashed lines light yellow, and blue).

## Cooccurrence of Cu-ATPases/CusAB and Cu-chaperones in the Cu-resistance operon

The first gene of this operon encodes Cu-ATPase CopA3, which is presumed to be a copper efflux pump (Fig. 3A, highlighted in yellow). Six hundred base pairs downstream of the *copA3* gene is encoded a cytoplasmic short cysteine-rich protein with a length of 110 amino acids and a four-helical copper-binding bundle that might have a role in copper storage or as a Cu chaperone capable of binding Cu(I) and delivering it to the inner membrane CopA which transports Cu(I) to the periplasm. One thousand base pairs downstream of genes coding Cu-binding proteins are encoded two transporters belonging to the resistance-nodulation and cell division (RND) superfamily. The first shares 70% identity with the inner membrane permease CusA from *E. coli*, and the second shares 80% similarity with the periplasmic membrane fusion protein from *E. coli* CusB. In enterobacteria, both proteins are part of a cation antiporter CusCFBA system for Cu(I) and Ag(I) detoxification (*Delmar, Su & Yu, 2015*). However, neither the outer membrane factor CusC nor the periplasmic metallochaperone CusF homologs were predicted by GN analysis 10 kb upstream and downstream of *cusAB* genes. The polypeptide of 73 amino acids encoded downstream *cusB* gene (Fig. 3A) contains a heavy metal-associated domain shared among Cu(I) chaperones; however, two-sequence BlastP comparisons did not identify significant alignments with CusF. Moreover, PSORT (https://www.psort.org/) and BUSCA (http://busca.biocomp.unibo.it/), two web servers for subcellular localization of proteins, could not predict a periplasmic localization characteristic of *E. coli* CusF. Only the protein WP_003603728.1 produced significant alignment (25.9% identity, 97% query cover and 2e-$^{23}$ e-value) with CusC from *E. coli* (WP_000074194.1). Both shared domains with outer membrane factors operate in conjunction with membrane fusion proteins to transport substrates across membranes. This putative *Mr. extorquens* CusC is part of a multidrug efflux system together with one CusB and two CusA proteins located far from the Cu-resistance operon.

## Putative Ni/Co resistance *rcnA/rcnR* operon

Downstream of the Cu-resistance operon, TREND predicted a couple of genes organized in a single operon with significant similarity (67%) to *E. coli* nickel/cobalt efflux proteins RcnA, and its transcriptional repressor, RcnR (Fig. 3A, light yellow dashed line).

### Pb/Cd/Zn homeostasis operon

An operon harboring seven genes (Fig. 3A, dark blue dashed line) is located downstream of the *rcnA/rcnR* operon in the complementary DNA strand. Thorough curation of the function of these proteins did not reveal significant information on their role in Cu homeostasis.

### Nucleases, CRISPR/Cas system, and mobility of Cu/Ni/Co-HGC

This multimetal homeostasis gene cluster is bordered at the 5′ end by a two-gene operon coding an exoDNAse V and a CRISPR/Cas system-associated protein Cas4 both sharing RecB domains (Fig. 3A, orange rectangles). The 3′ end is flanked by a type II endonuclease (orange rectangle), which is probably a remnant of the restriction-modification system (RM). It has been well documented that to maintain long-term persistence in the genome, RM and CRISPR/Cas systems are often linked with mobile genetic elements that confer an adaptive advantage. Their presence in the Cu/Ni/Co-HGC suggests that this gene cluster could be part of a genomic island prone to HGT (*Oliveira, Touchon & Rocha, 2014*).

### Cu homeostasis gene cluster-1 (Cu-HGC-1, 25.7 kb)

Figure 2 suggests a close phylogenetic relationship between CopA3 and CopA4. The GN of CopA4 revealed that this megaplasmid located Cu-HGC-1 and the chromosomal Cu/Ni/Co-HGC described above in the section "Gene neighborhoods analysis of Cu-ATPases uncovers a complex and potentially mobile Cu efflux resistome in *Mr. extorquens* AM1" share not only their Cu-ATPases but also the Cu-resistance and $cbb_3$-Cox biogenesis operons (Fig. 3B, green and light brown dashed lines).

TREND also predicted an extra copy of CusA and a fragment of CusB that might be the result of a genomic rearrangement involved in the duplication of this region. Cu-HGC-1 is flanked at the 5′ and 3′ ends by transposases belonging to the IS3, IS110, and IS630 families that resemble the structure of composite transposons. In addition, a protein with an integrase domain is also located at the 5′ end of Cu-HGC-1. The presence of these enzymes suggests the potential intra- and intergenomic mobility of this gene cluster.

### Megaplasmid-located Cu-homeostasis gene cluster-2 (Cu-HGC-2) encodes Cu efflux redundant functions

Figure 4A shows the gene neighborhood analysis of the Cu-ATPase CopA5. Genes located upstream and downstream of *copA5* encode the RND transporters CusB and CusA, and the Cu chaperones CopZ and CusF, respectively. These genes span 16 kb and together integrate into the second Cu homeostasis gene cluster (Cu-HGC-2). No genetic elements involved in mobility were detected.

### An unknown interrelation between efflux of copper and modulation of membrane charge highly conserved among the PPFM was found in the Cu-homeostatic gene cluster-3 (Cu-HGC-3)

The Cu-ATPase-encoding gene *copA1* is located in a chromosomal locus. According to TREND, *copA1* forms part of an operon together with two downstream genes (Fig. 4B).

The product of the first gene shows significant identity (58%) with CueR from *E. coli*, a Cu (I)-responsive transcriptional regulator protein. The second gene encodes a putative M20 family metallopeptidase. Since almost all naturally occurring metallopeptidases are zinc-dependent enzymes, the functional relationship between the products of these three genes could not be established. Upstream of the *copA*1 gene, TREND predicted a second two-gene operon. The first gene, *lpiA*, encodes a polypeptide of 343 amino acids with a significantly similar identity (31.16%) with the lysylphosphatidylglycerol synthetase from *R. tropici* CIAT899 (WP_015341009.1) responsible for lysylphosphatidylglycerol (LPG) formation. LPG produces a positive charged membrane that defends bacteria from cationic antimicrobial molecules. This cytoplasmic membrane lipid is synthesized only at low pH (4.5) and is involved in acid tolerance (*Sohlenkamp et al., 2007*). Downstream *lpiA*, is encoded a sensor kinase member of the PhoR family (WP_012753105) that may be involved in sensing phosphate status. The functional interrelation between *lpiA* and *copA* is unknown, it can be speculated the efflux of copper under low pH conditions. This genetic organization is not shared between the acid-tolerant strain *R. tropici* CIAT899 and *Mr. extorquens* AM1 but is highly conserved in the PPFM studied in this work.

## The mobile genetic element CopA2 (MGE-CopA2, 24.7 kb) shares characteristics of both integrative and conjugative elements (ICE), and composite transposons (CTn)

The gene encoding the Cu-ATPase CopA2, along with numerous genes often present in MGEs, are located in a chromosomal locus (Fig. 5). In the complementary DNA strand, upstream *copA2* gene is encoded a site-specific integrase that may catalyze the integration/excision of MGE. Downstream of the integrase gene, is encoded a type II toxin-antitoxin system (highlighted in blue), which is important for the long-term persistence of MGEs in their hosts (*van Melderen, de Bast & Rosenberg, 2009*). Downstream this T-A system is encoded an antirestriction protein homologous to ArdC (highlighted in gray), which may inhibit type I and type II R-M systems of the recipient cell and avoid degradation of the incoming MGE upon an HGT event (*González-Montes et al., 2020*). The presence of two IS3 family transposases bordering this genetic region (orange rectangles) resembles the structure of a composite transposon. The complexity of this putative MGE increases due to the presence of a *traG* gene at the 5′ end (highlighted in yellow). The product of this gene may be a component of the type IV secretory system essential for DNA transfer in bacterial conjugation. The relaxase domain-containing protein (highlighted in yellow) encoded four kb downstream of *traG* may initiate the conjugative transfer of DNA binding to the origin of transfer (*oriT*) and melt the double helix. The *copA2* gene, together with the integrative and conjugative transfer-associated functions described above, resembles the structure of a complex MGE known as an integrative and conjugative element (ICE). An ICE exhibits two different states: an integrative state, in which its DNA resides in the chromosome of the host, and a conjugative state, in which its DNA is excised from the chromosome of the host and can potentially spread horizontally by conjugative transfer to a new cell (*Delavat et al., 2017*).

## DISCUSSION

The antimicrobial property of copper has been considered a therapeutic alternative to control the dissemination of bacteria resistant to multiple antibiotics. The ability of bacteria to tolerate copper must be considered when implementing copper-based treatments, and a thorough understanding of Cu tolerance is required to achieve bactericidal efficacy with Cu therapies. Copper tolerance is the result of an evolutionary process initiated two billion years ago when the great oxidation event increased the bioavailability of this metal. In recent centuries, industrial and agricultural activities have spread Cu in different ecosystems. In contrast to other antimicrobials, metals cannot be degraded; thus, they have accumulated in the atmosphere, waters, and soils (*Briffa, Sinagra & Blundell, 2020*). These data suggest that environmental bacteria may be natural reservoirs of copper resistance genes similar to the genetic reservoir reported for antibiotic resistance genes (*Larsson & Flach, 2022*). Based on these data, high copper tolerance should be a widespread characteristic in bacteria and may interfere with the therapeutic use of copper.

No information is available on how copper tolerance is evolving in bacteria living in natural environments potentially exposed to neglected copper pollution. To contribute to this knowledge, we take advantage of a collection of models PAFD and PPFM constituted by root-nodulating and cosmopolitan freestyle bacteria respectively presumed to come from environments not polluted with metals. Their genomes, completely sequenced, allowed us to analyze not only their copper tolerance but also to infer their molecular mechanisms conferring resistance and involvement in HGT. With exception of *R. etli* CFN42 and *R. tropici* CIAT899 the copper tolerance of the PAFD and PPFM analyzed in this study have not been reported.

MIC comparisons indicate that strains isolated from nodules were much more sensitive to $CuCl_2$ than bacteria with a free lifestyle. Six of the eight PAFD analyzed are narrow-host range nitrogen-fixing symbiotic bacterium that exhibited the lowest range of tolerance between 0.020 and 0.030 mM. *S. fredii* NGR234 exhibited a MIC of 0.090 mM. This broad-host range nitrogen-fixing symbiotic bacterium is able to nodulate 79 legume plant genera (*Pueppke & Broughton, 1999*). Its ability to colonize different niches may have increased its exposure to copper and consequently its copper resistance.

The highest tolerance in the PAFD sample was found in *R. freirei* PRF81 (MIC 0.1 mM) an acid-tolerant strain able to grow and fix nitrogen at pH 4.8 (*Tullio et al., 2019*). A correlation between acid and metal tolerances has been observed in different bacteria. Cell-envelope modifications have been associated with low pH tolerance by reducing membrane permeability to $H^+$ (*Martinić et al., 2011*; *Ormeno-Orrillo et al., 2012*; *Shabala & Ross, 2008*). In extremely acidophilic bacteria with optimal growth at pH <3 and able to survive in metal-rich environments, changes in membrane permeability to $H^+$ has also been demonstrated to provide a barrier to metal influx (*Dopson et al., 2014*).

The PPFM sample exhibited higher copper tolerance than the PAFD sample, and their $CuCl_2$ tolerance ranged between 0.08 and 1.9 mM. In general, these bacteria are well adapted to a wide range of abiotic stresses such as toxic compounds, gamma and UV

radiation, heavy metals, chlorine, salinity, and desiccation (*Dourado et al., 2015*; *Dourado et al., 2012*). The highest CuCl₂ MIC (1.9 mM) was found in *Mr. extorquens* AM1 isolated in Oxford, England, in 1960 as an airborne contaminant in a medium containing methylamine as sole carbon and energy source. *Mr. extorquens* AM1 is a model for studying methylotrophy but there is no information on its copper resistance. To our knowledge, this is the first comprehensive genome-based inference study that elucidates the copper-efflux resistome in PPFM.

To gain insights into the evolution of copper tolerance in PAFD and PPFM, we analyzed the occurrence and diversity of their Cu-ATPases. The number of Cu-ATPases harbored in a single genome can vary between one and five. With the MIC data obtained in this study, we searched for a positive correlation between a high number of Cu-ATPases and a high MIC, but the data lack a correlation. From a mechanistic point of view, it can be explained by differences in turnover rates, transport studies have shown that Cu-ATPases with different efflux rates play different physiological roles distinct from resistance (*Raimunda et al., 2011*). Based on the evolution of virulence in opportunistic pathogens (*Brown, Cornforth & Mideo, 2012*; *Sheppard, 2022*), the lack of correlation between multiplicity and increased tolerance may be the result of a coincidental selection of Cu-ATPases with differences in turnover rates. A kind of preadaptation with no immediate environmental success but with long-term benefits.

The diversity of the Cu-ATPases harbored in PAFD and PPFM was assessed by phylogenetic analysis. Figure 2 shows that the multiple Cu-ATPases harbored in a sole strain are dispersed among distant lineages. This evolutionary divergence observed among several Cu-ATPases suggests that they may have been acquired by HGT. This phylogeny also revealed a putative duplication of Cu-ATPases MeAM3Ch and MeAM4Mp located in the chromosome and in the megaplasmid of *Mr. extorquens* AM1. It may represent an alternative mechanism to generate multiple Cu-resistance genes.

More data on the evolution of copper tolerance come from the thorough gene context analysis of the five Cu-ATPases encoded in the genome of *Mr. extorquens* AM1. This analysis revealed that genes coding for Cu-ATPases, copper efflux system CusAB and copper chaperones CusF and CopZ constitute putative Cu-homeostasis operons (Figs. 3A, 3B, and 4A). The multimetal Cu/Ni/Co/Zn homeostasis gene cluster shown in Fig. 3A is constituted by three different putative operons predicted by TREND based on distances between genes. The presence of genes coding putative integrases and transposases suggests that some of these Cu-HGCs may be transmitted intra- and intergenomically.

A similar genomic organization of putative copper-related genes in metal tolerance clusters was also found in environmental and clinical isolates of the opportunistic pathogen *A. baumannii* (*Thummeepak et al., 2020*; *Williams et al., 2016*).

Studies on antimicrobials (metals, antibiotics, and biocides) in the environment propose that the selection for copper resistance genes in the natural habitat of bacteria is positively influenced by the use of Cu-based pesticides and fertilizers in aquaculture and agriculture (*Singer et al., 2016*).

Numerous studies have demonstrated that several *Methylobacterium* and *Methylorubrum* species are nosocomial opportunistic pathogens, and have alerted on their

chlorine resistance, biofilm formation, desiccation tolerance, and high-temperature resistance, however, their copper tolerance has not been determined (*Cordovana et al., 2019*; *Kovaleva, Degener & van der Mei, 2014*; *Lai et al., 2011*; *Szwetkowski & Falkinham, 2020*). Our study suggests that *Mr. extorquens* AM1 could act as a reservoir of copper resistance genes prone to exchange genetic information which eventually may be connecting environmental with clinical resistance. Future analyses should be focused on examining the co-occurrence of copper and antibiotic resistance genes as well as if *Mr. extorquens* AM1 shares virulence traits in common with opportunistic pathogens. These data are required to provide clues on the emergence of this bacterium as a potential pathogen recalcitrant to copper-based treatments.

## CONCLUSIONS

The presence of bacteria with high copper tolerance is not restricted to metal-rich environments. This study showed that *Mr. extorquens* AM1 is an environmental isolate with high copper tolerance. Its inferred Cu efflux resistome suggests that this bacterium is well adapted to colonize and persist in niches with high copper content. The additive or synergistic effect of the five putative homeostasis gene clusters must be investigated to conclusively determine the contribution of the five putative homeostasis gene clusters to the high copper tolerance of *Mr. extorquens* AM1.

Our findings, together with the high copper tolerant strains reported in a large collection of environmental isolates of *A. baumannii* indicate that copper resistance genes are spreading in natural bacterial populations. However, more studies are required to generalize the presence of high copper-tolerant bacteria among environmental isolates and assess their interference with the therapeutic use of copper.

## ACKNOWLEDGEMENTS

The authors are grateful to Laura Cervantes, Karla Zeferino, and J. Pedro Elizalde-Díaz for their skillful technical assistance. Bacterial strains were kindly donated by Professor Esperanza Martínez-Romero (Centro de Ciencias Genómica, UNAM).

### Funding

This work was supported by UNAM-DGAPA-PAPIIT (Grant Number IN213619 to Alejandro García-de los Santos). The funders had no role in study design, data collection and analysis, decision to publish, or preparation of the manuscript.

### Grant Disclosures

The following grant information was disclosed by the authors:
UNAM-DGAPA-PAPIIT: IN213619.

### Competing Interests

The authors declare that they have no competing interests.

## Author Contributions

- Araceli Dávalos conceived and designed the experiments, performed the experiments, analyzed the data, authored or reviewed drafts of the article, and approved the final draft.
- Alejandro García-de los Santos conceived and designed the experiments, performed the experiments, analyzed the data, prepared figures and/or tables, authored or reviewed drafts of the article, and approved the final draft.

## Data Availability

The raw data is available in the Supplemental Files.

## Supplemental Information

Supplemental information for this article can be found online at http://dx.doi.org/10.7717/peerj.14925#supplemental-information.

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
