# Peer review of "Five copper homeostasis gene clusters encode the Cu-efflux resistome of the highly copper-tolerant Methylorubrum extorquens AM1"

_PeerJ, doi:10.7717/peerj.14925_

## Round 0.1 · original submission · Major Revisions

This manuscript has been evaluated by two reviewers, with different recommendations for the manuscript. Reviewer 1's critiques are valid, but I would be handle a resubmission of this manuscript that addresses this reviewers major points about placing the results in more thorough context as well as a broader discussion of copper in agriculture in the intro/discussion as per reviewer 2.

Reviewer 1 ·

Basic reporting

In their manuscript “Copper efflux resistome of the highly copper-tolerant Methylorubrum extorquens AM1 highlights the emergence of opportunistic pathogens resistant to copper-based treatments.” Davalos & Garcia-de los Santos describe the occurrence and genetic background of Cu-ATPases in Methylobacterium and Methylorubrum lab strains and their connection to increased copper tolerance.

The manuscript’s language and formal criteria are overall appropriate and the research question posed is of general interest. However, the chosen methodology as well as interpretation of results is lackluster and needs serious additional work:

Overall, the manuscript remains descriptive and speculative. It merely describes that certain ATPases were found in certain model strains, however, the results are not discussed or critically assessed, which results in no significant knowledge gain from the study in its current form. Further, the results are not put into a proper scientific context. This is especially exemplified when looking at the very short discussion and conclusion section, where the sole reference refers to the accumulation of copper in the environments.

In their description of the genes occurring in the vicinity of the ATPases regularly genes are exclusively named without any further description regarding their function. This would make the majority of the manuscript inaccessible for readers not familiar with the names of the copper resistance/tolerance genes.

Experimental design

In their title, introduction and discussion the authors connect their study to nosocomial infections. Their chosen strains however are plant-derived lab model organisms that are within the same genus, but might due to different habitat and niche specialization not have a lot in common with those facultative pathogenic strains that occur in the clinic. If this point is to be upheld, the relatedness of the chosen organisms to clinical pathogens as well as the potential occurrence of the described gene clusters within these pathogens need to be verified. Currently this point is not based on any evidence but pure speculation.

The authors state in their abstract, introduction and discussion that the complexity of the Cu efflux resistome is the result of a strong selection pressure exerted by copper. However, they similarly state that the strains originate from nonmetal-polluted environments. Which of the two is true?

Validity of the findings

The authors claim that the observed Cu ATPase clusters in Methylorubrum extorquens are directly linked to its far increased copper tolerance. However, they detect similar or even higher numbers of Cu ATPase clusters in several other of their model strains. As they state themselves, (eg. Line 272) no clear correlation between the abundance of such clusters and Cu tolerance can be established. Consequently, it is necessary to establish which of these clusters do indeed have a significant impact on copper tolerance through for example the creation of knock-out strains and subsequent copper tolerance analysis.

Additional comments

• mm is a bad choice as an abbreviation for minimal medium as it is commonly referring to millimeters

• the accession codes for the genomes of the strains should be give in the table 1

Reviewer 2 ·

Basic reporting

adequate

Experimental design

adequate

Validity of the findings

While the authors are generally cautious to not over-interpret, they stray across the line in a few places and need to soften their interpretations.

Line 120 "High copper tolerance is prevalent among PPFM"
based on other studies or this one only? If this study only 5 strains is not enough for a wide claim. If others, please cite.

Line 126: The complexity of this Cu efflux resistome is the result of a strong selection pressure exerted by copper in the environment.
This is an assumption

Line 370: This is over speculative
At most it would reasonable to speculate that this gene cluster may be induced under low pH, that's not enough to imply function or genetic requirement under that condition.

Line 436 The prevalence of low MIC values
exhibited by PAFD, even in strains with multiple Cu-ATPases, suggests that their plant hosts may contribute to protecting them against the toxic effects of copper.

Far too speculative to infer based on the small number of genomes with no strong basis of comparison

Additional comments

Consider copper is in wide and general use for the management of bacterial disease in agriculture, it would be appropriate to have some additional discussion on the current use of copper-based bactericides in agriculture, particularly since both of these groups of organisms commonly associate with plants.

---

## Round 0.2 · Minor Revisions

Thank you for addressing many of the comments of the reviewers. However, reviewer 1 still has some concerns about the way that some of the manuscript is framed, and these concerns need to be addressed prior to acceptance.

Reviewer 1 ·

Basic reporting

The revised manuscript “The copper efflux resistome of the highly copper-tolerant Methylorubrum extorquens AM1 highlights the emergence of opportunistic pathogens resistant to copper-based treatments.” has addressed a number of my (reviewer #1) previous comments well. In general, the manuscript has vastly improved compared to its previous version.

The discussion around long term selection for copper resistance in the introduction has removed one of the major weaknesses of the manuscript. The discussion now puts the study into a scientific context including references.

The authors have not provided any additional experimental proof regarding the annotation of copper resistance genes, which I can see based on their response could be considered outside the scope of this manuscript.

However, my major and most severe concern regarding speculation (see below) still remains and has not been adressed in any type of way.

Experimental design

No further comment

Validity of the findings

The entire study is still framed around mere speculation that M. extorquens might be an opportunistic pathogen and might have any clinical relevance. I don’t see any scientific proof that any of these claims are real. The authors mention it however in the title, the abstract (background as well as last sentence in results) and use the clinical importance throughout the introduction as well as discussion sections to frame their study. They do not show any pathogenicity genes encoded in the genome of the strain, they do not perform any genomic comparison with strains of clinical relevance.
Hence the study is currently still, either purposefully or neglectfully, framed in a way that misleads the reader. Indeed their response regarding my previous comment (see reviewer #1 experimental design) to the matter described above displays a severe misinterpretation of the study referenced by Sanz-Garcia et al.. The quoted paragraph in their response letter states that regularly clinical and environmental isolates are indistinguishable. However, the authors interpret this to imply that their specific environmental isolate can hence also be of clinical concern, which is simply not proven.
I agree with the authors that their study has merit as a purely descriptive study displaying “The copper efflux resistome of the highly copper-tolerant Methylorubrum extorquens AM1” however, the second part of their title “highlights the emergence of opportunistic pathogens resistant to copper-based treatments.” Is not supported in any scientific way through the contents of the manuscript.
In conclusion, either proper genomic proof for the pathogenic potential and relatedness of M. extorquens to isolates of clinical concern is needed, or the manuscript needs to be severely toned down to simply describe the copper efflux resistome without wide reaching implications based on pure speculation.

Additional comments

In addition it would be nice to add error bars to Figure 1 as it is based on replicates.

---

## Round 0.3 · Minor Revisions

Thank you for your resubmission. I think the manuscript is at a point where I do not need to send it out for re-review, but there are changes that need to be made based off of my reading and of the previous reviewer's comments that still must be made prior to acceptance.

I think that it is important to downplay the potential of this particular isolate (AM1) as a pathogen simply because it's a well worked on strain that has never really been shown to be a pathogen and thus any language otherwise will be viewed very skeptically by others in the field. Therefore, as you'll see, I think that most (and all if possible) mentions of the potential for this particular strain as a pathogen should be modified or deleted. Additionally, the analyses looking for particular pathogenesis genes just basically shows genes that are consistently present in other Gram negative species (whether pathogens or not) because of subjective definitions of "virulence factors". For instance, just because a strain has efflux pumps and motility genes does not make it a pathogen. Therefore, I would strongly suggest deleting these analyses from the manuscript and just focusing on the copper genes and resistance.

Here are my comments to be considered prior to resubmission:


Page 2: Last sentence, delete "presence of numerous putative virulence factors"

Page 2: Last sentence, "on the emergence of an opportunistic pathogen recalcitrant to copper-based treatments" better as "the presence of relatively high copper tolerance in environmental isolates of Methylorubrum"

Line 52: delete "that additionally harbors putative virulence factors"

Line 62: delete "survey for the presence of putative virulence factors"

Line 74: "on the emergence of an opportunistic pathogen recalcitrant to copper-based treatments" better as "the presence of relatively high copper tolerance in environmental isolates of Methylorubrum"

Line 157: delete "Numerous pathogenicity genes coding putative virulence factors"

Line 282-289: please take this analysis out of the manuscript

Line 461-478: please take this analysis out of the manuscript

Line 562: "AM1 is a reservoir" better as "could act as a reservoir"

---

## Round 0.4 · accepted · Accept

Thank you for addressing the prior comments and congratulations on the manuscript acceptance.